# Metabolite profiles associated with disease progression in influenza infection

**Chris H. Wendt**[1,2]*, **Sandra Castro-Pearson**[3], **Jennifer Proper**[3], **Sarah Pett**[4], **Timothy J. Griffin**[5], **Virginia Kan**[6], **Javier Carbone**[7], **Nikolaos Koulouris**[8], **Cavan Reilly**[3], **James D. Neaton**[3], **for the INSIGHT FLU003 Plus Study Group**[¶]

**1** Pulmonary, Allergy, Critical Care and Sleep Medicine Section, Minneapolis Veterans Administration Health Care System, Minneapolis, Minnesota, United States of America, **2** Division of Pulmonary, Allergy, Critical Care and Sleep Medicine, University of Minnesota, Minneapolis, Minnesota, United States of America, **3** Division of Biostatistics, University of Minnesota, Minneapolis, Minnesota, United States of America, **4** Medical Research Council Clinical Trials Unit, University College London, London, United Kingdom, **5** Department of Biochemistry, Molecular Biology, and Biophysics, University of Minnesota, Minneapolis, MN, United States of America, **6** Infectious Diseases Section, Veterans Administration Health Care System, and George Washington University, Washington, DC, United States of America, **7** Clinical Immunology Department, Hospital General Universitario Gregorio Maranon, Madrid, Spain, **8** Respiratory Medicine Dept, National and Kapodistrian University of Athens Medical School, Athens, Greece

¶ Membership of the INSIGHT FLU003 Plus Study Group is listed in the Acknowledgments.
* wendt005@umn.edu

## Abstract

### Background

We performed metabolomic profiling to identify metabolites that correlate with disease progression and death.

### Methods

We performed a study of adults hospitalized with Influenza A(H1N1)pdm09. Cases (n = 32) were defined by a composite outcome of death or transfer to the intensive care unit during the 60-day follow-up period. Controls (n = 64) were survivors who did not require transfer to the ICU. Four hundred and eight metabolites from eight families were measured on plasma sample at enrollment using a mass spectrometry based Biocrates platform. Conditional logistic regression was used to summarize the association of the individual metabolites and families with the composite outcome and its major two components.

### Results

The ten metabolites with the strongest association with disease progression belonged to five different metabolite families with sphingolipids being the most common. The acylcarnitines, glycerides, sphingolipids and biogenic metabolite families had the largest odds ratios based on the composite endpoint. The tryptophan odds ratio for the composite is largely associated with death (OR 17.33: 95% CI, 1.60–187.76).

**Data Availability Statement:** All relevant data are within the paper and its Supporting Information files.

**Funding:** This work was supported by a Subcontract award 13XS134 under Leidos

Biomed's Prime Contract [HHSN261200800001E and HHSN261201500003I], NCI/NIAID to The INSIGHT FLU 003 study awarded to JN. Protected research time was provided by the Department of Veterans Affairs Office of Research and Development to CW and VK. This work was also supported by a subcontract from University Minnesota [HHSN261200800001E and HHSN261201500003I], NCI/NIAID and protected time from the University of New South Wales awarded to SP.

**Competing interests:** I have read the journal's policy and the authors of this manuscript have the following competing interests: The views expressed in this article are those of the authors and do not reflect the views of the US Government, the National Institutes of Health, the Department of Veterans Affairs, the funders, or any of the authors' affiliated academic institutions. This does not alter our adherence to PLOS ONE policies on sharing data and materials.

## Conclusions

Individuals that develop disease progression when infected with Influenza H1N1 have a metabolite signature that differs from survivors. Low levels of tryptophan had a strong association with death.

## Registry

ClinicalTrials.gov Identifier: NCT01056185

## Introduction

The INSIGHT Influenza Hospitalization study (FLU 003) is an international observational cohort study that was launched in 2009 to characterize A(H1N1)pdm09 infection. Previous studies from this cohort identified baseline elevations of biomarkers associated with inflammation, coagulation and/or immune function as predictors for disease progression [1]. In addition, for the same cases and controls considered in this investigation, we previously carried out a targeted analysis for 2 specific metabolites, tryptophan (T) and kynurenine (K) and found significantly elevated KT ratio among cases, consistent with tryptophan catabolism, and found an elevated KT ratio was associated with worse clinical outcomes following hospitalization [2]. Tryptophan catabolism is also reported in influenza associated encephalopathy where metabolic profiling identified additional metabolite biomarkers [3]. These findings motivated an investigation of a larger number of metabolites using our same case:control design.

Metabolomics is the systematic identification, quantification and characterization of metabolites, the products of metabolism, within an organism or biological sample. Metabolomics has emerged as a useful tool to identify biomarkers of disease and identify putative pathways of disease [4]. Influenza infection is a systemic infection with broad physiological ramifications. These ramifications include the metabolome, which is perturbed in both animal and cellular models of influenza infection [5, 6]. Recent studies, including our own, have demonstrated perturbations in the metabolome in influenza infection in humans [2, 7]. In addition to tryptophan catabolism, broad perturbations that include metabolite families such as purines, pyrimidines, acylcarnitines, fatty acids, amino acids, glucocorticoids, sphingolipids, and phospholipids are found in animal models of influenza pneumonia [8]. Using nuclear magnetic resonance technology, human studies have identified alterations in amino acids, sugars and other small molecules in influenza associated with lung injury and pneumonia [9, 10]. NMR has the advantage of using a targeted approach; however, it has limitations in metabolite profiling [4]. In this study, we used a targeted, quantitative mass spectrometry-based approach to measure 408 metabolites across several metabolome families and identify metabolites associated with poor clinical outcomes, death or transfer to intensive care in patients hospitalized for influenza A(H1N1)pdm09 infection.

## Methods

### Study design and objectives

FLU 003 is an ongoing, international observational study of adults hospitalized with influenza that began in 2009 following the pandemic infection with the influenza A(H1N1)pdm09 virus (ClinicalTrials.gov Identifier: NCT01056185). This was a matched nested case-control study whose results were previously reported [2]. Cases (n = 32) were FLU 003 patients with PCR-

confirmed influenza A(H1N1)pdm09 virus with a poor outcome following hospitalization defined as a composite outcome of death or transfer from the general ward to the intensive care unit (ICU) during the 60-day follow-up period after enrollment. Controls (n = 64) had PCR-confirmed influenza A(H1N1)pdm09 virus, survived the 60-day follow-up period, were not transferred to the ICU, and were matched on age (+/- 4 years) and gender. The objective of this exploratory study was to determine whether metabolic profiling would identify metabolomic families and/or specific metabolites that differed between those with a poor outcome (cases) compared to controls.

### Ethics statement

The FLU 003 protocol and information statement and consent form were approved by both the local institutional ethics committees/review boards of the participant sites and the ethics committee of the Sponsor of this study, the University of Minnesota. All participants or their representatives (when participants were unable to consent for themselves, and where the ethics permission allowed consent by a third party) provided written informed consent prior to their enrollment.

### Mass Spectrometry (MS) analysis

At study enrollment, blood was drawn into EDTA tubes and plasma was processed within 4 hours as previously described [2]. All plasma samples used in this analysis had undergone two freeze-thaw cycles. For metabolite identification, 10μl of plasma was manually loaded onto a Biocrates Life Sciences Absolute IDQ p400 HR (Biocrates Life Sciences catalog number 21018) following the manufacturer's instructions. Analysis was performed on a Thermo Scientific, Q Exactive TM, Hybrid Quadrupole-Orbitrap TM, mass spectrometer equipped with a Thermo Scientific Ultimate 3000 UHPLC equipped with an autosampler. Sample metabolite quantification was performed with the integrated MetIDQ Biocrates software [11]. The Biocrates platform contains standards for eight families of metabolites for a total of 408 individual metabolites. Internal controls are incorporated for normalization between plates. The limit of detection (LOD) for each metabolite is provided by the Biocrates manufacturer and is calculated by Met/DQ$^{TM}$ and is defined as three times the background noise level. Families (number of metabolites) measured included: acylcarnitines (55), amino acids (21), biogenic amines (21), monosaccharide (1), di- and tri- glycerides (60), phospholipids (lysophosphatidylcholines and phosphatidylcholines) (196), sphingolipids (ceramides and sphingomyelins) (40) and cholesteryl esters (14) (S1 Table in S1 Appendix).

### Statistical methods

This study used the same cases and controls from our previous study [2]. Descriptive statistics were used to summarize the baseline characteristics of cases and controls. Metabolites with values below LOD in both the case and control groups that were present in 10% or more subjects were removed from consideration prior to analysis (S1 Table in S1 Appendix). Values that fell below the LOD for the remaining 188 metabolites were imputed with half of their corresponding LOD. One metabolite was further removed from analysis due to lack of variability that precluded the creation of tertiles. Conditional logistic regression that accounted for the matching by age and gender was used to summarize the association of the metabolite families and 187 individual metabolites with the composite outcome, death and transfer to the ICU. All models were adjusted for duration of symptoms at enrollment, which was significantly associated with the composite outcome in univariate analyses of potential confounding factors.

For analyses by metabolite family, a new covariate was created by summing the standardized values of metabolites, which had mean 0 and standard deviation 1 across all subjects, from the same family. If individual metabolites in a family were associated with the composite outcome in a similar manner, we reasoned that such an analysis would provide improved power compared to a study of the individual metabolite. The analyses by family also provide a means of controlling for type 1 error.

As a first step, we investigated kynurenine and tryptophan concentrations that were previously reported but with different laboratory methods [2]. Next we studied metabolite families and individual metabolites and focused our discussion on associations with p<0.01 to provide some control of type 1 error. P-values cited are based on models that use continuous variables for the metabolite or family covariates. Odds ratios cited compare the upper and lower tertiles and 95% confidence intervals (CIs) are given.

## Results

Thirty-two participants met our case definition, of whom 22 died and 10 required transfer to the ICU during the follow-up period. Two controls were available for all cases that were matched for sex and did not differ significantly for race, smoking status or presence of underlying lung disease. In addition to matching factors, since influenza is a respiratory illness, we considered two potential confounders for disease progression: days since onset of influenza symptoms and history of lung disease (COPD and/or asthma) at the time of enrollment. Cases had been symptomatic for a median of eight days, whereas controls had been symptomatic for a median of six days (p = 0.04 for the difference) (Table 1). Furthermore, 19% and 22% of cases and controls reported lung disease at time of enrollment, respectively (p = 0.70 for the difference) (Table 1). Duration of symptoms was also significantly associated with the composite outcome and therefore included in subsequent conditional logistic regression analyses.

In a previous publication we reported tryptophan and kynurenine concentrations using different methods, specifically single reaction monitoring with MS/MS [2]. For this analysis using the Biocrates platform with metabolite standards we performed adjusted conditional logistic regressions for kynurenine, tryptophan, and the KT ratio, the latter as a surrogate for tryptophan catabolism, to validate our previous findings (Table 2). This analysis was then repeated after dividing the data into fatal and nonfatal cases, along with their matched controls, to determine if there were any associations with the mortality component of the composite outcome (Table 2). For comparison purposes, we display the inverse odds ratio for tryptophan,

**Table 1. Clinical characteristics.**

| | Case (n = 32) | Control (n = 64) | |
|---|---|---|---|
| | No. (%) or Median (25th,75th %) | No. (%) or Median (25th,75th %) | p-value[b] |
| **Female[a]** | 13 (41) | 26 (41) | - |
| **Age[a]** | 52 (41, 60) | 53 (40, 60) | - |
| **Non-white race** | 7 (22) | 13 (20) | 0.84 |
| **Smoker** | 10 (36) | 22 (34) | 0.86 |
| **Days since onset of influenza symptoms** | 8 (6, 10) | 6 (4, 7) | 0.04 |
| **Lung Disease** | 6 (19) | 14 (22) | 0.70 |

[a]Matching factor
[b]Univariate conditional logistic

**Table 2. Conditional logistic regression results for kynurenine, tryptophan, and the KT ratio.**

|  | Composite Endpoint | | | Mortality Endpoint | | | ICU Transfer Endpoint | | |
|---|---|---|---|---|---|---|---|---|---|
|  | OR | 95% CI | P-value[3] | OR | 95% CI | P-value[3] | OR | 95% CI | P-value[3] |
| KYN[1] | 2.96 | 0.91–9.60 | 0.005 | 3.11 | 0.73–13.37 | 0.008 | 3.17 | 0.35–28.45 | 0.513 |
| TRP[2] | 3.34 | 0.91–12.23 | 0.032 | 17.33 | 1.60–187.76 | 0.013 | 0.21 | 0.02–2.70 | 0.356 |
| KT Ratio[1] | 2.61 | 0.81–8.39 | 0.010 | 3.11 | 0.73–13.37 | 0.014 | 1.86 | 0.25–13.85 | 0.580 |

[1]Tertile 3 vs. Tertile 1 Odds Ratio

[2]Tertile 1 vs. Tertile 3 Odds Ratio

[3]P-values obtained from models with continuous covariates

i.e., those comparing the lower tertile to the upper tertile. The odds ratios (cases vs. controls) for kynurenine, tryptophan, and the KT ratio are, respectively, 2.96 (95% CI, 0.91–9.60), 3.34 (95% CI, 0.91–12.23), and 2.61 (95% CI, 0.81–8.39).

When restricted to either fatal or nonfatal cases, the odds ratios for kynurenine and the KT ratio were similar to the odds ratios for the composite endpoint. In contrast, the tryptophan odds ratio for the composite endpoint appears to be largely determined by the death component. New for this analysis, we found among fatal and nonfatal cases the tryptophan odds ratios were 17.33 (95% CI, 1.60–187.76) and 0.21 (95% CI, 0.02–2.70), respectively. A graphical depiction of the relationships between mortality and the kynurenine and tryptophan tertiles can be seen in S1 Fig in S1 Appendix. Although these results are relatively imprecise due to the limited sample size, there is a strong negative association between mortality and the tryptophan tertiles consistent with our previous findings.

The Biocrates platform contains metabolites from eight major metabolite families. Table 3 displays the results for the adjusted conditional logistic regression by metabolite family for the composite endpoint and the restricted fatal and nonfatal datasets. The acylcarnitines, glycerides, sphingolipids, and biogenic amines had the strongest odds ratios in the analysis based on the composite endpoint. For the acylcarnitines and glycerides, the odds ratios for disease progression were 3.99 (95% CI, 1.03–15.42) and 3.69 (95% CI, 1.08–12.61), respectively. Because these odds ratios did not change substantially when restricted to either fatal or nonfatal cases, the simplification of the composite endpoint did not appear to have much influence on the odds ratios for these two families.

Acylcarnitines are also known to be associated with both insulin resistance and sepsis. We therefore evaluated the association of diabetes and sepsis with the composite outcome and acylcarnitine levels. The presence of diabetes was found to be associated with the composite

**Table 3. Conditional logistic regression results by metabolite family after adjusting for duration of symptoms.**

|  | Composite Endpoint | | | Mortality Endpoint | | | ICU Transfer Endpoint | | |
|---|---|---|---|---|---|---|---|---|---|
|  | OR | 95% CI | P-value[3] | OR | 95% CI | P-value[3] | OR | 95% CI | P-value[3] |
| Acylcarnitines | 3.99 | 1.03–15.42 | 0.009 | 4.99 | 0.77–32.25 | 0.010 | 3.08 | 0.28–34.34 | 0.546 |
| Amino Acids | 1.08 | 0.35–3.34 | 0.016 | 0.49 | 0.11–2.10 | 0.053 | 5.38 | 0.63–46.24 | 0.104 |
| Biogenic Amines | 2.21 | 0.73–6.68 | 0.007 | 1.94 | 0.49–7.70 | 0.009 | 3.60 | 0.43–30.35 | 0.603 |
| Phospholipids | 0.77 | 0.25–2.40 | 0.076 | 0.98 | 0.23–4.30 | 0.126 | 0.56 | 0.08–3.65 | 0.465 |
| Sphingolipids | 0.27 | 0.07–1.13 | 0.016 | 0.07 | 0.01–0.83 | 0.021 | 1.10 | 0.13–9.26 | 0.551 |
| Cholesterol Esters | 0.51 | 0.14–1.86 | 0.054 | 0.25 | 0.04–1.66 | 0.073 | 1.02 | 0.10–10.20 | 0.606 |
| Glycerides | 3.69 | 1.08–12.61 | 0.020 | 3.84 | 0.80–18.47 | 0.039 | 4.00 | 0.48–33.03 | 0.497 |

[1] P-values obtained from models with continuous covariates

**Table 4. Ten strongest odds ratios comparing the upper vs. lower tertile after adjusting for duration of symptoms and matching factors.**

| Metabolite | Odds Ratio (OR) | 95% CI for OR | P-value[1] | Inverse Odds Ratio | Family |
|---|---|---|---|---|---|
| TG.48.3. | 10.79 | 2.11–55.27 | 0.296 | 0.09 | Glycerides |
| SM.38.1. | 0.10 | 0.02–0.63 | 0.006 | 10.00 | Sphingolipids |
| AC.2.0. | 9.26 | 1.96–43.73 | 0.008 | 0.11 | Acylcarnitines |
| PC.O.38.6. | 0.14 | 0.04–0.55 | 0.019 | 7.14 | Phospholipids |
| SM.33.2. | 0.15 | 0.03–0.70 | 0.299 | 6.67 | Sphingolipids |
| TG.54.4. | 6.53 | 1.34–31.90 | 0.128 | 0.15 | Glycerides |
| Phe | 6.45 | 1.52–27.44 | 0.006 | 0.16 | Amino Acids |
| LPC.O.18.1. | 0.16 | 0.04–0.66 | 0.003 | 6.25 | Phospholipids |
| SM.37.1. | 0.16 | 0.04–0.67 | 0.037 | 6.25 | Sphingolipids |
| SM.41.1. | 0.16 | 0.04–0.76 | 0.007 | 6.25 | Sphingolipids |

[1]P-values obtained from models with continuous covariates. TG = triglyceride, SM = sphingomyelin, AC = acylcarnitine, PC = phosphatidylcholine,

Phe = phenylalanine, LPC = lysophosphatidylcholine

outcome (OR, 5.58; 95% CI, 1.16–36.10; p = 0.014) whereas this was not observed with sepsis (OR, 2.09; 95% CI, 0.26–16.54; p = 0.397). A Wilcoxon rank sum test found no significant differences in acylcarnitine levels based on diabetes (median difference, -0.285; 95% CI, -5.48–4.97; p = 0.95).

The odds ratios varied for the sphingolipids and biogenic amine metabolite families. While the odds ratio for the composite endpoint for the biogenic amines was 2.21 (95% CI, 0.73–6.68), this value decreased to 1.94 (95% CI, 0.49–7.70) for fatal cases and notably increased to 3.60 (95% CI, 0.43–30.35) for nonfatal cases. Similarly, while the odds ratio for the composite endpoint for the sphingolipids was 0.27 (95% CI, 0.07–1.13; inverse OR, 3.70), this value decreased among fatal cases (OR, 0.07; 95% CI, 0.01–0.83; inverse OR, 14.29) and increased among nonfatal cases (OR, 1.10; 95% CI, 0.13–9.26; inverse OR, 0.98). As TNF-α can stimulate sphingolipid levels [12, 13], we performed a Wilcoxon rank sum test for our case:control study and found no significant differences in TNF-α levels based on the composite endpoint (median difference, 1.940; 95% CI, -0.300–4.270; p = 0.078); however, there was an association when restricted to mortality (median difference, 3.810; 95% CI, 0.860–7.860; p = 0.011). TNF-α levels were also found to be associated with sphingolipids with a Pearson's correlation coefficient of -0.205 (95% CI, -0.390- -0.005; p = 0.045).

Table 4 shows the results for the adjusted conditional logistic regression for the 187 individual metabolites included in the analysis. The odds ratios comparing those in the upper tertile to those in the lower tertile are displayed for the ten metabolites found to have the strongest association with the composite outcome. Inverse odds ratios are provided so that metabolites with odds ratios above and below one can be effectively compared. For instance, the adjusted odds ratio comparing those in the upper vs. lower tertile for the triglyceride TG.48.3 and sphingomyelin SM.38.1 were, respectively, 10.79 (95% CI, 2.11–55.27) and 0.10 (95% CI, 0.02–0.63). Given the inverse odds ratio for SM.38.1 comparing those in the lower vs. upper tertile is 10.00, we note that TG.48.3 has a slightly stronger association with case-control status than SM.38.1.

Notably, several of the metabolites in Table 4 are from the same family. Four of these metabolites, for example, are in the sphingolipid family, which is notable since only 10% of the analyzed metabolites are from the sphingolipid family (S1 Table in S1 Appendix). Remaining metabolites include two each from the glyceride and phospholipid families. Fig 1 demonstrates the log odds ratios of the 187 analyzed metabolites separated by metabolite family. The

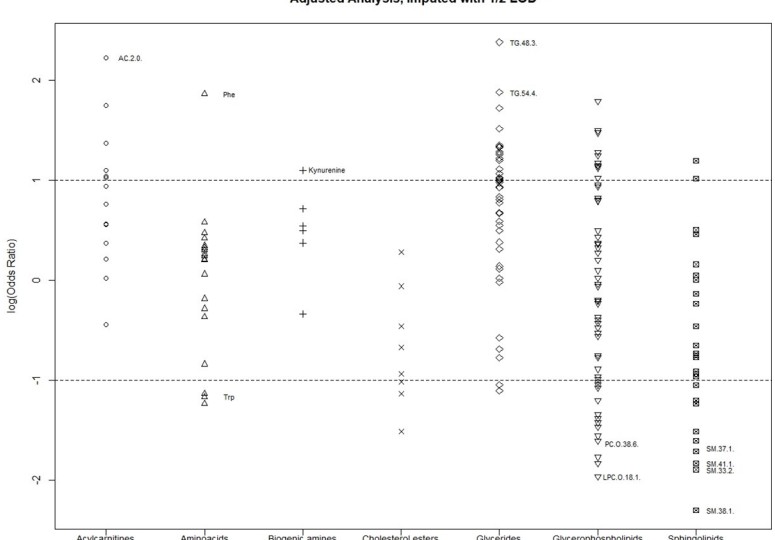

**Fig 1. Log odds ratios comparing the upper vs. lower tertile after adjusting for duration of symptoms and matching factors by metabolite family.** Referenced: Ten strongest odd ratios as well as Kynurenine and Tryptophan.

metabolites listed in Table 4 along with kynurenine and tryptophan are labeled in the plot. Values below 0 suggest that subjects with lower metabolite values have a greater chance of being a case. In particular, the vast majority of acylcarnitines, biogenic amines, and glycerides have positive log odds ratios. This suggests that metabolites from these families have similar associations with case-control status.

## Discussion

Infection with influenza is a major public health concern and currently there is no clear test or biomarker to identify individuals at risk for disease progression, such as respiratory failure or death. We have previously identified a strong association between metabolites involved in tryptophan metabolism and disease progression [2]. We have now extended these studies to include a broader metabolomic profile with biomarkers and metabolite families that are associated with disease progression.

In this study we found a number a metabolomic families including individual metabolites associated with disease progression in influenza infection. We also noted that several of these metabolites were from the same family of metabolites. Amongst these families acylcarnitines, glycerides, sphingolipids, and biogenic amines had the strongest association based on our composite endpoint of death and/or respiratory failure.

Acylcarnitines belong to a family of metabolites involved in fatty acid transport and certain plasma carnitines are elevated in insulin resistance [14]. Plasma acylcarnitines are also elevated in sepsis and have been shown to predict outcome. Specifically, Green and colleagues found that plasma acylcarnitines at the time of sepsis diagnosis differentiated survivors from non-survivors [15]. In our study, we found that diabetes was associated with the composite outcome of death or transfer to the ICU but not with acylcarnitine levels. The occurrence of sepsis was not associated with the composite outcome or acylcarnitine levels. Therefore, it is unknown if the acylcarnitine levels are unique to influenza infection and warrants further study.

Sphingolipids were also altered in our subjects with severe influenza infection. Sphingolipids not only serve as structural components of the plasma membrane lipid bilayer but also participate in cell signaling. Much of what we know about sphingolipid signaling comes with the advent of advanced mass spectroscopy techniques that allow the simultaneous analysis and quantification of multiple sphingolipid species, such as we used in this study. Sphingolipid metabolites play key roles in immune cell migration and function [16, 17] and have been associated with sepsis and poor outcomes [18]. In addition, the pro-inflammatory cytokine TNF-α stimulates sphingolipid metabolism [12, 13]. In our INSIGHT cohort we reported elevated levels of TNF-α associated with disease progression following H1N1 infection [1] and in this subgroup we found an association of TNF-α with sphingolipid levels. In a ferret model of H1N1 respiratory tract infection the sphingolipid sphingomyelin correlated with viral titers [5]. Viral titers were unavailable for this analysis; therefore, we are unable to determine whether sphingolipid metabolism correlates with influenza titers. However, our finding of sphingolipid metabolism appears to primarily associate with progression to critical illness.

We previously reported in this group that tryptophan metabolism is associated with disease progression as reflected by an increase in the kynurenine/tryptophan ratio [2]. In this current study our metabolomic profiling confirmed these previous measurements. When we focused on fatal cases, we found that the odds of death for low tryptophan levels were almost 26-fold higher. Induction of tryptophan metabolism has been demonstrated in both animal models and human infection with influenza [7, 19]. Similar to our findings, tryptophan and its main metabolic pathway have been associated with poor outcomes in inflammatory and infectious diseases [20–22].

Lastly, we sought to determine if lung disease, such as COPD or asthma, was a confounder for tryptophan and its metabolite kynurenine. Viral infection, including influenza, is a common cause of COPD exacerbation and those with COPD have demonstrated worse outcomes with H1N1 infection [23, 24]. In addition, tryptophan metabolism through the kynurenine pathway is associated with COPD exacerbations [25, 26]. However, in our small study we did not find lung disease to be a confounder for either the tryptophan metabolites or disease progression.

## Conclusion

In summary, a strength of this study is the demonstration of a metabolomic signature that associates with progression to death or respiratory failure in a relatively small case:control study of adults hospitalized with influenza A(H1N1)pdm09. This signature is enriched for metabolites with known associations to critical illness and poor outcomes. The results agree with our previous work [5] insofar as clear associations were found between kynurenine and the KT ratio and disease progression. In addition, low tryptophan levels are associated with a very high likelihood of death among those hospitalized for influenza. While this study has identified several classes of metabolites associated with poor outcome in the setting of A (H1N1)pdm09 infection, it is limited by the relatively small sample size as reflected by some large confidence intervals for some metabolites. Future studies could benefit from validating these findings in larger cohorts, other types/subtypes of influenza infection and include longitudinal testing to determine the durability of these signals.

## Supporting information

**S1 Appendix.**
(DOCX)

## Acknowledgments

We extend our grateful thanks to all the volunteers who have been participating in this study. INSIGHT **International coordinating centers: Copenhagen:** Bitten Aagaard, Álvaro H. D. Borges, Alessandro CozziLepri, Marius Eid, Per O. Jansson, Marianne Jeppesen, Zillah Maria Joensen, Ruth Kjærgaard Pedersen, Jens Lundgren, Birgit Riis Nielsen, Mary Pearson, Lars Peters, Tavs Qvist. **London:** Brian Angus, Abdel Babiker, Rachel Bennett, Nafisah Braimah, Yolanda Collaco-Moraes, Adam Cursley, Fleur Hudson, Sarah Pett, Charlotte Russell, Helen Webb. **Sydney:** Dianne Carey, David Courtney-Rodgers, Sean Emery, Pamela Shaw. **Washington:** Fred Gordin, Adriana Sanchez, Barbara Standridge, Michael Vjecha. **Statistical and Data Management Center, Minneapolis, Minnesota:** Kate Brekke, Megan Campbell, Eileen Denning, Alain DuChene, Nicole Engen, Michelle George, Merrie Harrison, James D. Neaton, Ray Nelson, Siu-Fun Quan, Terri Schultz, Deborah Wentworth. **Specimen repositories and laboratories:** John Baxter, Shawn Brown (Leidos Biomedical Research, Inc.), Marie Hoover (ABML). **National Institute of Allergy and Infectious Disease/Leidos:** John Beigel, Richard T. Davey Jr., Robin Dewar, Erin Gover, Rose McConnell, Julia Metcalf, Ven Natarajan, Tauseef Rehman, Jocelyn Voell.**Institute of Clinical Pathology and Medical Research, NSW Health Pathology, Westmead Hospital and University of Sydney, Westmead, New South Wales, Australia:** Dominic E. Dwyer, Jen Kok. **Centers for Disease Control and Prevention, Atlanta, Georgia:** Timothy M. Uyeki. **Community representative:** David Munroe.

   **Clinical site investigators by country: Argentina**: Damian Aguila, Maria Fernanda Alzogaray, Maria Fernanda Ballesteros, Laura Barcan, Laura Barcelona, Waldo Belloso, Veronica Berdiñas, Pablo Bonvehi, Juan Pablo Caeiro, Veronica Cisneros, Ana Crinejo, Daniel David, Luz Doldan, Juan Ebenrstejin, Flavio Lipari, Ana Lopardo, Gustavo Lopardo, Marcelo Losso, Pablo Lucchetti, Sergio Lupo, Laura Moreno Macias, Alejandra Moricz de Tesco, Analia Mykietiuk, Estaban Nannini, Gabriel Nieto, Laura Nieto, Luciana Peroni, Ignacio Retta, Patricia Rodriguez, Marisa Sanchez, Pablo Sanchez, Mariana de Paz Sierra, Silvina Tavella, Elena Temporiti, Liliana Trape, Ines Vieni, Eduardo Warley, Diego Yahni, Abel Humberto Zarate. **Thailand**: Anchalee Avihingsanon, Kanlaya Charoentonpuban, Ploenchan Chetchotisakd, Peeraporn Kaewon, Naphassanant Laopraynak, Weerawat Manosuthi, Kanitta Pussadee, Opass Putcharoen, Kiat Ruxrungtham, Gompol Suwanpimonkul, Sasiwimol Ubolyam. **United States**: Roberto Arduino, Barbara Atkinson, Taryn M. Aulicino, Jason V. Baker, Cindy Bardascino, Caitlin Bass, John D. Baxter, Mark Beilke, Beverly D. Bentley, Mary Lee Bertrand, Ann B. Brown, June Carbonneau, Richard Cindrich, Patty Coburn, Calvin J. Cohen, Linda Clark, Shirley Cummins, Paul Dassow, Jack A. DeHovitz, Nila J. Dharan, Leslie Faber, Marti Farrough, Matthew Freiberg, Edward Gardner, Kimberly Jo Garrett, Christiane Geisler, Marshall Glesby, Julia Green, Joanne Grenade, Edie Gunderson, John Gunter, Kirsis Ham, Susan Holman, Valery Hughes, Christopher Hurt, Mary Johnson, Glory Koerbel, Susan Koletar, Audrey Lan, Rodger MacArthur, Cheryl Marcus, Norm Markowitz, Maria Laura Martinez, Karen McLaughlin, Raquel Nahra, Mary Jane Nettles, Daniel Nixon, Richard Novak, Kathleen Nuffer, Hannah B. Olivet, Bola Omotosho, Armando P. Paez, Marta Paez-Quinde, Sonija Parker, Namrata Patil, Hari Polenakovik, Sandra Powell, Rachel A. Prosser, Nancy A. Reilly, Paul F. Riska, Stacey Rizza, Robert Schooley, Marla Schwarber, James Scott, Gary L. Simon, Jon Sivoravong, Daniel J. Skiest, Clemencia Solorzano, Rita Sondengam, Nicole Swanson, Ellen Tedaldi, Zelalem Temesgen, Doug Thomas, Bill Thron, Colleen Traverse, David E. Uddin, Daniel Z. Uslan, Marina Vasco, William M. Vaughan, Isabel Vecino, Barbara Wade, Catrice Walker, Kathy Watson, Vicky Watson, David Wohl, Cameron R. Wolfe. **Belgium**: Leslie Andry, Mireille Bielen, Nathan Clumeck, Eric Florence, Kabamba Kabeya, Jolanthe Sagaer, Jozef Weckx. **Greece**: Olga Anagnostou, Anastasia Antoniadou, George Daikos, Vicky Gioukari, Ioannis

Kalomenidis, Maria Kantzanou, Georgios Koratzanis, Nikolaos Koulouris, Efstratios Maltezos, Symeon Metallidis, Vlassis Polixronopoulos, Helen Sambatakou, Athanasios Skoutelis, Giota Touloumi, Nikolaos Vasilopoulos. **Australia**: Mark Bloch, Nicky Cunningham, Dominic E. Dwyer, Sian Edwards, Julian Elliott, Jill Garlick, Philip Habel, Fiona Kilkenny, Helen Lau, Karen MacRae, John McBride, Richard Moore, Isabel Prone, Ristila Ram, Sue Richmond, Norm Roth, Tuck Meng Soo, Jo-Anne Thompson, Trina Vincent, Emanuel Vlakahis, Rachel Woolstencroft. **United Kingdom**: David Chadwick, Tristan Clarke, Jane Democratis, David Dockrell, Robert Heyderman, Ben Jeffs, Stefan Kutter, Martin Llewelyn, Jane Minton, Melanie Newport, Ashley Price. **Peru**: Carlos Benites, Raul Castillo, Romina Chinchay, Eva Cornelio, Maria Guevara, Luis Gutierrez, Jose Hidalgo, Alberto La Rosa, Yvett Pinedo, Maria Saenz, Juan Vega. **Denmark**: Bente Baadegaard, Karen Bach, Philippa Collins, Jan Gerstoft, Lene Hergens, Lene Pors Jensen, Zillah Maria Joensen, Gitte Kronborg, Iben Rose Loftheim, Henrik Nielsen, Lars Oestergaard, Court Pedersen, Jens Aage Stauning, Svend Stenvang Pedersen, Yordanos Yehdego. **Germany**: Frank Bergmann, Christoph Boesecke, Johannes R. Bogner, Norbert Brockmeyer, Christine Czaja-Harder, Rika Draenert, Gerd Fätkenheuer, Hartwig Klinker, Tim Kümmerle, Clara Lehmann, Vera Müller, Andreas Plettenberg, Jürgen Rockstroh, Stefan Schlabe, Wolfgang E. Schmidt, Dirk Schürmann, Gundolf Schüttfort, Ulrich Seybold, Christoph Stephan, Albrecht Stoehr, Klaus Tillmann, Susanne Wiebecke, Timo Wolf. **Spain**: Jose Arribas, Javier Carbone, Eduardo Fernández Cruz, David Dalmau, Vincente Estrada, Patricia Herrero, Hernando Knobel, Paco López, Rocío Montejano, José Sans Moreno, José Ramón Paño, Begoña Portas, Maria Rodrigo, Pilar Romero, Domingo Sánchez-Sendín, Vincente Soriano. **Poland**: Elzbieta Bakowska, Andrzej Jerzy Horban, Brygida Knysz, Karolina Pyziak Kowalska, Anna Zubkiewicz-Zarebska. **Estonia**: Kerstin Kase, Helen Mülle, Kai Zilmer. **Chile**: Gladys Allendes, Jimena Flores, Rebeka Northland, Carlos Perez, Isabel Velasco, Marcelo Wolff. **China**: Man-Yee Chu, Tak-chiu Wu. **Austria**: Heinz Burgmann, Selma Tobudic. **Japan**: Mayumi Imahashi, Junji Imamura, Yasumasa Iwatani, Ayumi Kogure, Masashi Nakahata, Wataru Sugiura, Yoshiyuki Yokomaku. **Norway**: Anne Maagaard.

## Author Contributions

**Conceptualization:** Chris H. Wendt, Sarah Pett, James D. Neaton.

**Data curation:** Chris H. Wendt, Cavan Reilly, James D. Neaton.

**Formal analysis:** Chris H. Wendt, Sandra Castro-Pearson, Jennifer Proper, Cavan Reilly, James D. Neaton.

**Funding acquisition:** Sarah Pett, James D. Neaton.

**Methodology:** Chris H. Wendt, Timothy J. Griffin, Cavan Reilly.

**Project administration:** Chris H. Wendt.

**Supervision:** Chris H. Wendt.

**Writing – original draft:** Chris H. Wendt, Sandra Castro-Pearson, Jennifer Proper.

**Writing – review & editing:** Sandra Castro-Pearson, Jennifer Proper, Sarah Pett, Timothy J. Griffin, Virginia Kan, Javier Carbone, Nikolaos Koulouris, Cavan Reilly, James D. Neaton.

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
