## [Decision Letter · Decision Letter 0]

9 Feb 2021

Metabolite Profiles associated with Disease Progression in Influenza Infection

PONE-D-20-32204

Dear Dr. Chris,

We’re pleased to inform you that your manuscript has been judged scientifically suitable for publication and will be formally accepted for publication once it meets all outstanding technical requirements.

Kind regards,

Ch Ratnasekhar, Ph.D.

Academic Editor

PLOS ONE

Journal Requirements:

1.Thank you for stating the following in the Competing Interests section:

"I have read the journal's policy and the authors of this manuscript have the following competing interests: The views expressed in this article are those of the authors and do not reflect the views of the US Government, the National Institutes of Health, the Department of Veterans Affairs, the funders, or any of the authors’ affiliated academic institutions."

Please respond by return email with your amended Competing Interests Statement and we will change the online submission form on your behalf.

Additional Editor Comments (optional): I would suggest to submit the raw data in online public access platform like Metabolights or NIH public accession data sumbmission

Reviewer comments

Thanks for the opportunity to review this manuscript. I would like to submit my recommendation directly via email (as given below).

My response is as follows -

1. Recommend (Accept)

2. Is the manuscript technically sound, and do the data support the conclusions? (YES)

3. Has the statistical analysis been performed appropriately and rigorously? (Yes)

4. Does the manuscript adhere to the PLOS Data Policy? (Yes) Expect for deposit of mass spectrometry data which I have recommended in my comments

5. Is the manuscript presented in an intelligible fashion and written in standard English? (Yes)

6. Review Comments to the Author: (Pasted below)

7. Would you like your identity revealed to the authors of this submission? (Yes)

8. Do you have any potentially competing interests? None

Review Comments to the Author:

In the manuscript titled as "Metabolite Profiles associated with Disease Progression in Influenza Infection” the authors report the identification of plasma metabolites that may be correlated with influenza disease progression. The cohort on which this study has been carried out is part of the INSIGHT Influenza Hospitalization study (FLU 003) which is focused on characterising the A(H1N1)pdm09 infection. Prior work as part of this study has revealed biomarkers as predictors of disease progression. In fact, as report by the authors, the same cohort samples used in this study have been analysed earlier for 2 metabolites - tryptophan (T) and kynurenine (K). The previous reported a significantly elevated KT ratio among cases whose clinical outcomes worsened following hospitalisation. The present study has been undertaken by the authors to followup on the earlier study and to find out if any other metabolites have any correlation with worsening disease outcome.

In this study, the authors have used a quantitative Metabolomics approach to measure hundreds of metabolites to identify those associated with death or ICU treatment in patients hospitalised for influenza A(H1N1)pdm09 infection. The manuscript reports that the metabolite signature in disease survivors is different from that of those who develop severe disease. Particularly, the authors report that low levels of tryptophan is strongly associated with death. The study is designed and although the number of patients included in the study is small (small cohort size), the conclusion made are statistically significant.

The Biocrates methodology and standards have been used in this study to rigorously analyse the metabolites from plasma samples. This is one of the best available technologies for conducting the quantitative metabolomics work undertaken. First, the authors were able to confirm their previous finding that low levels of tryptophan is strongly associated with death; mortality end point - OR 17.33 (95% CI, 1.60-187.76) & ICU end point - OR 0.21 (95% CI, 0.02-2.70), respectively. The study also reports that two other groups of metabolites - acylcarnitines & glycerides - which exhibited significant association with composite end point (both death & ICU treatment), with OR 3.99 (95% CI, 1.03-15.42) pvalue 0.009 & OR 3.69 (95% CI, 1.08-12.61) pvalue 0.020, respectively. However, they are not predictors of mortality. Other groups of metabolites such as biogenic amines and sphingolipids are reported to have better correlation with non-fatal cases.

Further analysis on individual metabolites identified 10 that were significantly associated with composite end point. Interestingly, these metabolites are mainly form the sphingolipid, glyceride and phospholipid families. In summary, the authors conclude that acylcarnitines, glycerides, sphingolipids, and biogenic amines have the strongest association with the composite endpoint in this study. Primarily this study has helped to confirm and establish earlier findings that decreased tryptophan levels are highly correlated with death due to influenza.

This study has demonstrated unequivocally that plasma metabolite signatures are valid biomarkers for disease progression in case of A(H1N1)pdm09 infection. Moreover this study also assumes significance in view of the current COVID19 pandemic, as identifying patients whose disease outcomes are likely to worsen is very important. The methodology laid out by this study is broadly applicable.

One important point is that the authors will have to deposit the mass spec data generated from this study in a public repository (such as Metabolomics work bench) and provide the data deposit ID in the methods section.

No corrections or changes recommended by this reviewer.
---

## [Editor Report · Acceptance letter]

25 Mar 2021

PONE-D-20-32204 

Metabolite Profiles associated with Disease Progression in Influenza Infection 

Dear Dr. Wendt:

I'm pleased to inform you that your manuscript has been deemed suitable for publication in PLOS ONE. Congratulations! Your manuscript is now with our production department. 

Kind regards, 

on behalf of

Dr. Ch Ratnasekhar 

Academic Editor

PLOS ONE